# Halide Perovskite glues activate two-dimensional covalent organic framework crystallites for selective NO$_2$ sensing

Wen Ye[1], Liangdan Zhao[2], Hong-Zhen Lin [3], Lifeng Ding[2], Qiang Cao[4], Ze-Kun Chen[4], Jia Wang[4], Qi-Meng Sun[4], Jing-Hui He[4] & Jian-Mei Lu [1,4] ✉

Two-dimensional covalent organic frameworks (2D COFs) are promising for gas sensing owing to the large surface area, abundant active sites, and their semiconducting nature. However, 2D COFs are usually produced in the form of insoluble micro-crystallites. Their poor contacts between grain boundaries severely suppress the conductivity, which are too low for chemresistive gas sensing. Here, we demonstrate that halide perovskites can be employed as electric glues to bond 2D COF crystallites to improve their conductivity by two orders of magnitude, activating them to detect NO$_2$ with high selectivity and sensitivity. Resonant microcantilever, grand canonical Monte Carlo, density functional theory and sum-frequency generation analyses prove that 2D COFs can enrich and transfer electrons to NO$_2$ molecules, leading to increased device conductivity. This work provides a facile approach for improving the conductivity of polycrystalline 2D COF films and may expand their applications in semiconductor devices, such as sensors, resistors, memristors and field-emission transistors.

Two-dimensional covalent organic frameworks (2D COFs) are promising for resistance-related applications, such as gas sensing, optoelectronic devices and photoelectric catalysis, owing to their semiconducting nature, large surface area, size-tunable pores and abundant active sites[1–5]. Among them, gas sensing plays an indispensable role in various fields, such as food quality assessment, plant growth detection, and noninvasive medical diagnosis[6–8]. To be compatible with the circuit of a sensing device, proper base resistances are needed[9]. Therefore, a great deal of effort has been devoted to tuning the conductivity of 2D COFs through molecular formula innovation, including monomers, linkages, and defects[10]. However, 2D COFs usually grow in the form of insoluble polycrystalline powders, where the poor contacts in grain boundaries severely inhibit the macroscopic conductivity of COFs[9]. Guest molecule doping aims to increase the carrier concentration in COF molecules within grains, the access to COFs with high bulk conductivity remains limited, and the manufacturing process is difficult[11,12]. Alternately, the preparation of continuous films or single crystals of COFs is also rather challenging and consequently cost-ineffective for potentially scalable applications[13–16]. Therefore, it is more practical to find an effective technique that can improve the contact resistances of COF crystallites for direct integration into sensors.

Halide perovskites have risen in recent years as superstar semiconducting materials in solar cells, photodetectors, electrical storage, light-emitting diodes, lasers, and many other applications[17–22]. Benefiting from their defect-insensitive resistances, halide perovskites can be prepared by a solvent method at room temperature, which is compatible with thermally susceptible COFs[23–26]. The metal sites exposed by these defects can coordinate with ligands from 2D COFs.

[1]State Key Laboratory of Radiation Medicine and Protection, Soochow University, Suzhou, China. [2]Department of Chemistry, Xi'an Jiao Tong-Liverpool University, Suzhou, China. [3]Department i-LAB, Suzhou Institute of Nano-Tech and Nano-Bionics (SINANO), Chinese Academy of Sciences, Suzhou, China. [4]College of Chemistry Chemical Engineering and Materials Science, Collaborative Innovation Center of Suzhou Nano Science and Technology, National United Engineering Laboratory of Functionalized Environmental Adsorption Materials, Soochow University, Suzhou, China. ✉e-mail: lujm@suda.edu.cn

Therefore, halide perovskites can be a suitable conductive glue for electrically linking COF crystallites.

Herein, we demonstrate that halide perovskites can act as a semiconductive glue to link 2D COF crystallites (Fig. 1a) to activate them for selective $NO_2$ sensing. High selectivity of the sensor was demonstrated with a response to $NO_2$ that was 70 times higher than that to 20 other gases of the same concentration, and high sensitivity with the lowest detection limit of 40 ppb was also exhibited. Resonant microcantilever (RMC), grand canonical Monte Carlo (GCMC), density functional theory (DFT), and sum frequency generation (SFG) analyses prove that 2D COFs can enrich and transfer electrons to $NO_2$ molecules, leading to increased device conductivity. Halide perovskites lower the boundary resistance between the COF crystallites without short-circuiting, in sequence modulating the entire sensor's basis conductance suitable for gas sensing. Our work gives a simple and effective approach to improve the conductivity of 2D COF films and expand the application of 2D COF crystallites in resistance-related applications.

## Results

### Structure and characterization

The synthesis of halide perovskite $Cs_2PdBr_6$ was reported in our previous work[27–29]. $PdBr_2$ and CsBr were dissolved in hydrobromic acid (HBr) at a molar ratio, the oxidant dimethyl sulfoxide (DMSO) was added, and the mixture was heated and stirred to form the double perovskite $Cs_2PdBr_6$ with a regular octahedral structure (Supplementary Fig. 1a). High-resolution transmission electron microscopy (HRTEM) and X-ray diffraction (XRD) characterizations demonstrate the synthesis of $Cs_2PdBr_6$ (Supplementary Figs. 1b and 2)[29]. Five 2D COFs were also synthesized by Schiff base reactions (Supplementary Figs. 3 and 4; see the "Methods" section for details)[30]. In addition, we successfully used perovskite nanospheres to link 2D COF crystallites by an anti-solvent growth method. First, TpPa-1 powders are dispersed in tert-butanol by sonification, and then the $Cs_2PdBr_6$ solution is added dropwise to the dispersion and fully stirred to form TpPa-1/$Cs_2PdBr_6$ (Supplementary Fig. 5). The XRD pattern showed that TpPa-1/$Cs_2PdBr_6$ retains the $Cs_2PdBr_6$ and TpPa-1 phases after ultrasonication (Supplementary Fig. 1c). The binding between $Cs_2PdBr_6$ and TpPa-1 was first investigated by XPS. In Fig. 1b–d, the $Pd^{4+}$ peak in TpPa-1/$Cs_2PdBr_6$ blue-shift compared to $Cs_2PdBr_6$, while the $Pd^{2+}$ peak did not shift. In addition, both the O peak and the N peak are red-shift compared to TpPa-1. These results indicate that TpPa-1 and $Cs_2PdBr_6$ form the simultaneous coordination of $Pd^{4+}$ with O on the carbonyl group and N on the imine group in TpPa-1 (Fig. 1i). $Pd^{4+}$ is formed due to the Br vacancy ($V_{Br}$) and Pd–Br antisite ($Pd_{Br}$) point defects inherent in perovskites[31]. The Fourier transform infrared (FT-IR) pattern also shows that the C–N bond in TpPa-1/$Cs_2PdBr_6$ has a significant shift compared to that in TpPa-1, which also shows that the N in the imine group is coordinated with Pd (Supplementary Fig. 4f). Field emission scanning electron microscopy (FSEM) and energy dispersive X-Ray spectroscopy (EDX) show that the perovskites are distributed on the surface of the COFs in the form of nanospheres so that the COF crystallites can be well connected through the perovskites (Fig. 1e–h). To demonstrate the role of the electric glue, the current–time plots of TpPa-1 and TpPa-1/$Cs_2PdBr_6$ under constant voltage were measured. We fabricated TpPa-1/$Cs_2PdBr_6$ films as the semiconducting active layer in sensor devices. The TpPa-1/$Cs_2PdBr_6$ film (~100 μm) was directly drop-coated on an $Al_2O_3$ substrate printed with interdigitated electrodes (channel width: 200 μm). With the addition of electric glue, the resistivity of the material decreased from $9.86 \times 10^{11}$ to $7.06 \times 10^9$ Ω m when a constant voltage of 5 V was applied across the interdigitated electrodes (Supplementary Fig. 6).

### Nitrogen dioxide sensing

Gas sensors play an indispensable role in various fields, such as food quality assessment, plant growth detection, and noninvasive medical diagnosis[6–8]. A chemresistive gas sensor with proper basis conductance is required to detect gas varying from one part per billion (ppb) to one part per million (ppm) among dozens or even hundreds of interfering gases (e.g., >800 in breath or >250 in indoor air)[32,33]. We dropped the TpPa-1/$Cs_2PdBr_6$ (51.6 wt%) sample on an interdigital electrode to fabricate chemresistors and tested their performance (Supplementary Figs. 5 and 7). Under a constant bias voltage of 5 V, the sensor responds to the flushing of $NO_2$ flow with increasing concentration (Fig. 2a). The TpPa-1/$Cs_2PdBr_6$ exhibits superiority in chemresistive detection of $NO_2$ with a detection limit of 40 ppb, which is lower than that of most $NO_2$ sensors (Supplementary Fig. 8). Even under an atmosphere with different humidity, the sensor can still work smoothly (Supplementary Fig. 9). Multiple tests of the same device have established the sensor's excellent reusability (Supplementary Fig. 10). At a concentration of 10 ppm $NO_2$, the response/recovery time was 71/254 s (Supplementary Fig. 11). The voltage–current characteristics remain in the ohmic mode, indicating that the current is mainly determined by the intrinsic conductivity of the film contributed by the thermal release of the carriers[34]. With increasing $NO_2$ concentration, the intrinsic conductivity of the film gradually increases (Fig. 2b). A high selectivity of the sensor's response to $NO_2$ is 70 times or more sensitive than that to the other 20 gases (CO, HCl, $NH_3$, NO, $SO_2$, $H_2$, acetone, 3-pentanone, ethyl acetate, butyl acetate, toluene, chlorobenzene, benzaldehyde, anisole, isopropanol, ethanol, n-heptane, n-hexane, acetic acid, and formic acid. Fig 2c). Stability is also an important parameter of sensor performance. After 160 days in the atmosphere, the sensor is still able to distinguish between different $NO_2$ concentrations with a response that is >60 at 2 ppm $NO_2$ (Fig. 2d).

### Universality of electric glue

To demonstrate the general nature of electric glue in the improvement of sensor performance, different COF/perovskite sensors were prepared and tested, including sensors for different $Cs_2PdBr_6$ contents, different COFs, and different halide perovskites. The addition of different contents of $Cs_2PdBr_6$ enhanced the response and selectivity of 2D COFs to $NO_2$. The results showed that, with the decrease in the amount of doping, the response and selectivity to $NO_2$ showed a trend of first increasing and then decreasing, and the incorporation of 51.6 wt% $Cs_2PdBr_6$ could maximize the sensor sensitivity (Fig. 3a). We incorporated 51.6% wt $Cs_2PdBr_6$ into five different 2D COFs (TpPa-1, TpPa-2, TpPa-CN, TpPa-$NO_2$, and TpPa-COOH). All these sensors have the most significant increase in response and selectivity to $NO_2$ (Fig. 3b). However, sensors based on COFs and perovskites alone have poor sensing performance and cannot meet the actual requirements (Fig. 3a, b and Supplementary Fig. 12). The combination of high mobility and low initial carrier concentration is critical in the development of ultrasensitive sensors for oxidizing gases[35]. We applied this method to other halide double perovskites ($Cs_2AgBiBr_6$, $Cs_3Bi_2Br_9$, and $Cs_2SnI_6$; halide double perovskites have better environmental stability than other $ABX_3$ halide perovskites) and proved good universality for double perovskites (Fig. 3c and Supplementary Fig. 13). XPS result indicates that TpPa-1 and the halide double perovskites form the chelation of metal ions (Ag, Bi, and Sn) by O from the carbonyl group and by N on the imine group in TpPa-1 (Supplementary Fig. 14).

### Mechanism

To understand the significant increase in sensor response and selectivity, we conducted an RMC test, which is able to detect the adsorbed mass of gaseous molecules (Supplementary Fig. 15). Figure 4f shows the prepared cantilever where the material is loaded onto its free end. Thereafter, the fabricated cantilever is placed inside the testing chamber for gas-sensing performance evaluation. With TpPa-1/

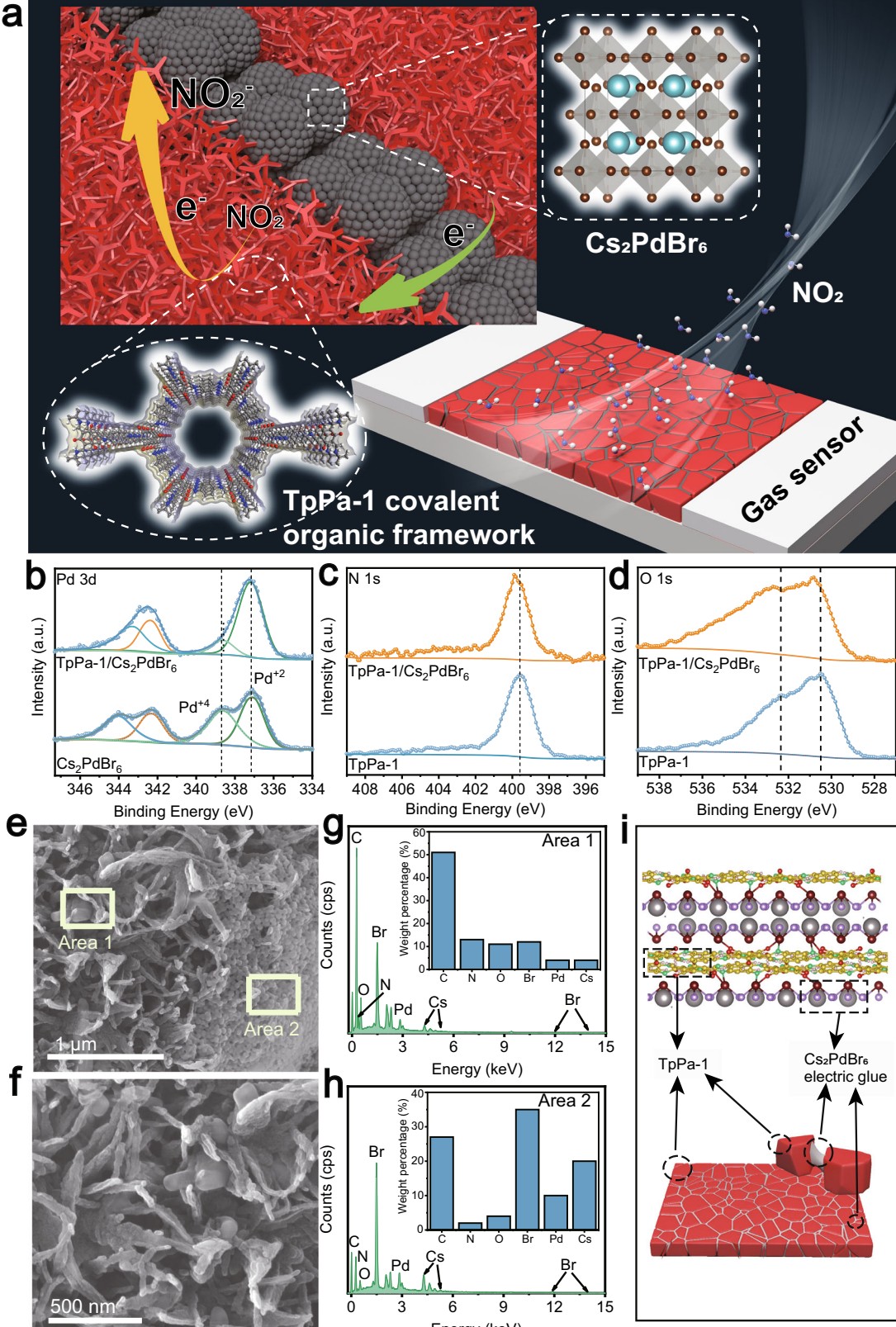

**Fig. 1 | Structure and characterization of electric glues bond 2D COF crystallites. a** Schematic diagram of the gas sensor and perovskite glue to bond COF crystallites. **b** XPS spectra of the Pd $3d$ region of $Cs_2PdBr_6$ and TpPa-1/$Cs_2PdBr_6$; **c** N $1s$ and **d** O $1s$ region of the TpPa-1 and TpPa-1/$Cs_2PdBr_6$. **e, f** FSEM images and **g, h** EDX analysis images of TpPa-1/$Cs_2PdBr_6$. **i** Theoretical models of the combination of Pd exposed by surface defects of perovskites and N and O in TpPa-1. The yellow, pink, red, green, violet, brown, and gray balls refer to C, H, O, N, Br, Pd, and Cs atoms, respectively. Source data are provided as a Source Data file.

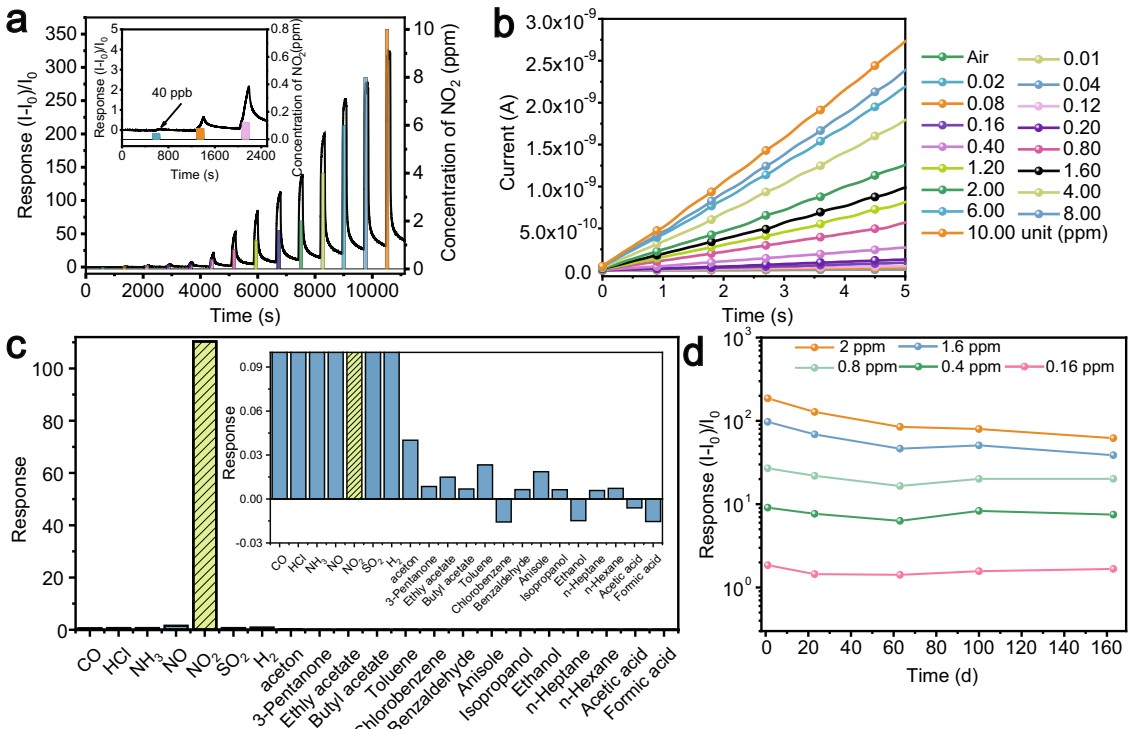

**Fig. 2 | Characterization of sensing performance. a** Responses variation versus $NO_2$ concentrations (40 ppb to 10 ppm). **b** Current–voltage characteristics for sensor when exposed to different $NO_2$ concentrations (0–10 ppm). **c** Gas selectivity of the TpPa-1/$Cs_2PdBr_6$ sensor. (Air is the carrier gas, and all gas concentrations are 2 ppm.) **d** The long-term stability of the TpPa-1/$Cs_2PdBr_6$ sensor toward different concentrations of $NO_2$. Source data are provided as a Source Data file.

$Cs_2PdBr_6$ deposited on resonant microcantilevers, the adsorbed $NO_2$ acts as an added mass to shift the cantilever resonant frequency for gravimetric sensing signal readout (Fig. 4b)[36]. The advantage of the RMC test is that the signal is only related to the adsorption of the gas, and the electron transfer between the gas molecules and the sensing material does not affect the test results, which is very important for the analysis of the sensing mechanism. As shown in Fig. 4b, the incorporation of COFs can greatly improve the gas adsorption capacity of the material, which is caused by the porous structure of COFs. For the convenience of comparison, we calculated the weight of gas adsorbed per nanogram of material (Fig. 4a). Figure 4a shows that COFs have a strong gas enrichment effect, which can partly explain the stronger response of TpPa-1/$Cs_2PdBr_6$ compared to $Cs_2PdBr_6$. The GCMC simulation proved that the average excess uptake of the $NO_2$/$N_2$ mixture was over 40 pg/ng of TpPa-1 or other COFs (Fig. 5a and Supplementary Figs. 16–20). However, this enrichment has no gas selectivity.

To further understand the sensing mechanism, we used DFT and SFG analyses of the adsorption of $NO_2$ molecules on TpPa-1/$Cs_2PdBr_6$, $Cs_2PdBr_6$, and TpPa-1 films. As shown in Fig. 5c, the SFG signal intensity of the three types of membranes significantly increases after treatment with $NO_2$, indicating that charge transfer occurs after $NO_2$ adsorption. The TpPa-1/$Cs_2PdBr_6$ film showed the greatest enhancement of the background signal, which indicates that $NO_2$ produces a stronger charge transfer between $NO_2$ and TpPa-1/$Cs_2PdBr_6$. Therefore, the TpPa-1/ $Cs_2PdBr_6$ film has an excellent response to $NO_2$. We initially propose 12 adsorption models based on the fact that 2D COFs have layered and porous structures and that $NO_2$ molecules typically adsorb on the surface and within the pores (Supplementary Fig. 21 and Supplementary Table 1). The DFT calculation results show that $NO_2$ is physically adsorbed on TpPa-1 (Supplementary Figs. 22 and 23). In addition, when the $NO_2$ molecule is close to TpPa-1/$Cs_2PdBr_6$, it will coordinate with Pd on the

perovskites (Supplementary Fig. 24). Therefore, the TpPa-1/ $Cs_2PdBr_6$ film presents a distinct characteristic peak of $NO_2$ after adsorption, which indicates $NO_2$ has a relatively high orientation uniformity and selectivity at room temperature (Fig. 5c)[37,38]. Our analysis shows that charge transfer between TpPa-1 and $NO_2$ molecules and the bond of $Cs_2PdBr_6$ glue to TpPa-1 crystallites significantly increase the base conductivity of the TpPa-1 film, which gives the sensor excellent sensing performance.

## Discussion

We proposed using halide perovskites as electric glues to bond 2D COF crystallites. These electric glues significantly improve 2D COF crystallite conductivity by two orders of magnitude, activating them to detect $NO_2$ with high selectivity and sensitivity. The combination of high mobility and low initial carrier concentration is critical in the development of ultrasensitive $NO_2$ sensors. The TpPa-1/$Cs_2PdBr_6$ sensor realized high selectivity (the sensor's response to $NO_2$ is 70 times or more sensitive than that to the other 20 gases) and high sensitivity (the lowest detection limit can reach 40 ppb) in the detection of $NO_2$. Overall, this work takes a simple and effective approach and has important implications for improving the conductivity of 2D COFs and advancing their resistance-related applications.

## Methods

### Synthesis of $Cs_2PdBr_6$

0.426 g CsBr (2 mmol, TCI, 99% purity), 0.266 g $PdBr_2$ (1 mmol, Aladdin, 99% purity) and 5 mL of 48% HBr (Macklin) were added to the three-necked flask, and the solution was heated to 85 °C with stirring for 5 min. The solution was continued to be heated. 0.5 mL of dimethyl sulfoxide (Macklin, AR) was added to it when the solution temperature reached 120 °C. In order to fully react, the solution was continued to stir for 10 min. After the solution

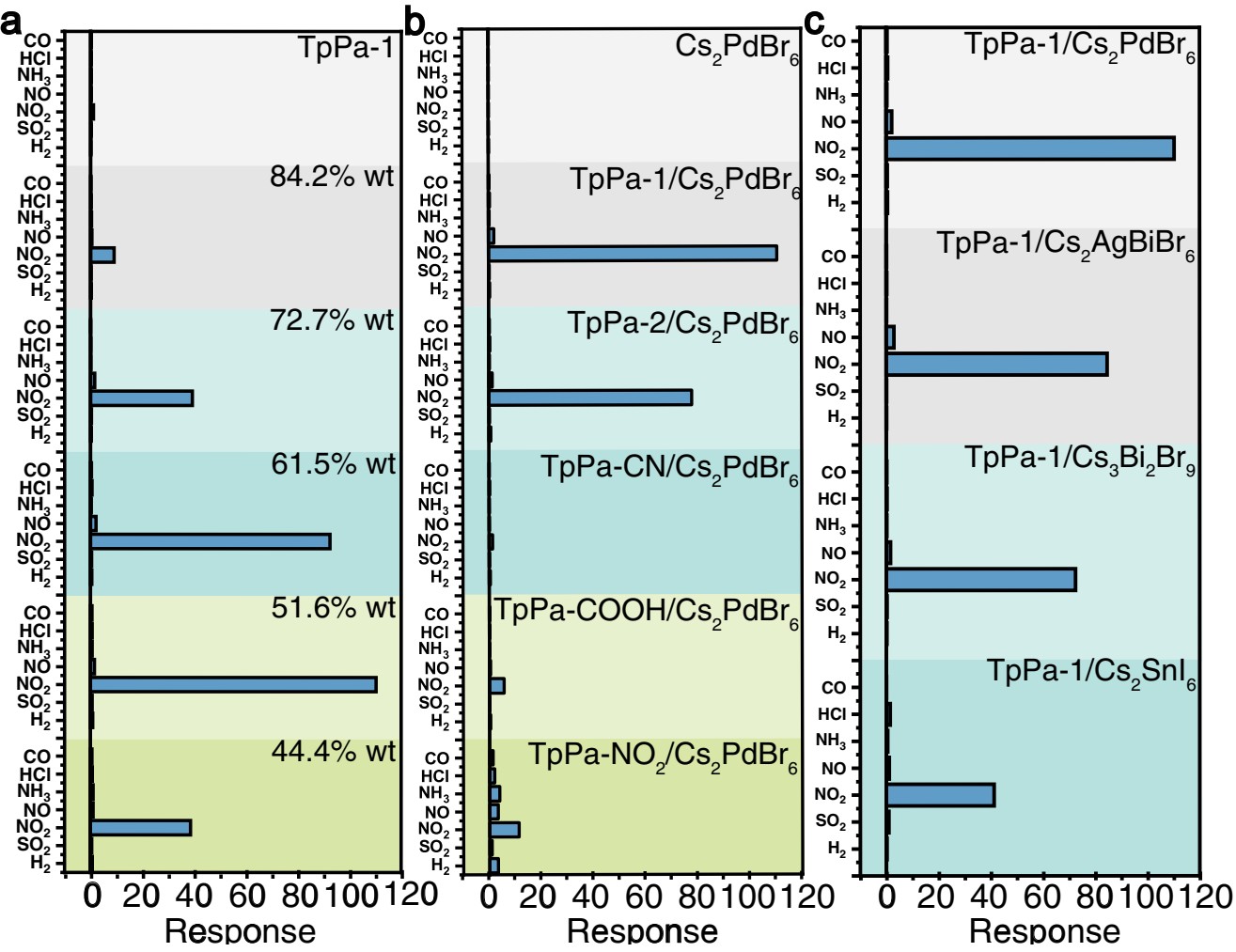

**Fig. 3 | Comparison of sensor gas response and selectivity performance (all gas concentrations are 2 ppm). a** The response of TpPa-1 to different gases after mixing different proportions of $Cs_2PdBr_6$. **b** The response of different COFs to different gases after mixing 51.6% wt $Cs_2PdBr_6$. **c** The response of TpPa-1 to different gases after mixing different perovskites. Source data are provided as a Source Data file.

cooled to room temperature, the solution containing the black $Cs_2PdBr_6$ crystalline was filtered and washed several times with toluene. The product was dried under reduced pressure at 100 °C overnight.

### Synthesis of $Cs_2AgBiBr_6$

$Cs_2AgBiBr_6$ was successfully synthesized according to the reported method[39]. 0.426 g CsBr (2 mmol, TCI, 99% purity), 0.449 g $BiBr_3$ (1 mmol, TCI, 99% purity), and 0.188 g AgBr (1 mmol, TCI, 99% purity) were mixed with 10 mL of 48% HBr in a round-bottomed flask. The solution was continuously stirred for 2 h at 120 °C. The solution was allowed to stand for 2 h after cooling to room temperature to obtain an orange precipitate. Subsequently, the solution containing the orange $Cs_2AgBiBr_6$ crystalline was filtered and washed several times with ethanol. The product was dried under reduced pressure at 100 °C overnight.

### Synthesis of $Cs_3Bi_2Br_9$

$Cs_3Bi_2Br_9$ was successfully synthesized according to the reported method[40]. 0.638 g CsBr (3 mmol), 0.897 g $BiBr_3$ (2 mmol), and 5 mL of 48% HBr were added to the three-necked flask. The solution was continuously stirred at 80 °C, heated for 1 h, and

then cooled to room temperature. The solution containing yellow $Cs_3Bi_2Br_9$ powder was filtered and washed several times with ethanol. The product was dried under reduced pressure at 100 °C overnight.

### Synthesis of $Cs_2SnI_6$

$Cs_2SnI_6$ was successfully synthesized according to the reported method[41]. 3.258 g $Cs_2CO_3$ (10 mmol, Aladdin, 99% purity) was mixed with 20 mL of 55% HI (Macklin) in a 100 mL beaker to afford a concentrated acidic solution of CsI. 3.132 g $SnI_4$ (5 mmol, Aladdin, 99% purity) was dissolved in 10 mL ethanol to afford a clear orange solution. The $SnI_4$ solution was added to the CsI solution under stirring, and black solids were continuously precipitated. In order to fully react, the solution was continued to stir for 10 min. The solution containing Black $Cs_2SnI_6$ powder was filtered and washed several times with ethanol. The product was dried under reduced pressure at 100 °C overnight.

### Synthesis of TpPa-1 COF

126 mg triformylphloroglucinol (Tp) (0.6 mmol, Macklin, 97% purity), 96 mg p-phenylenediamine (Pa-1) (0.9 mmol, Aladdin, 99% purity), 16.5 mL of mesitylene (TCI, 97% purity), and 16.5 mL

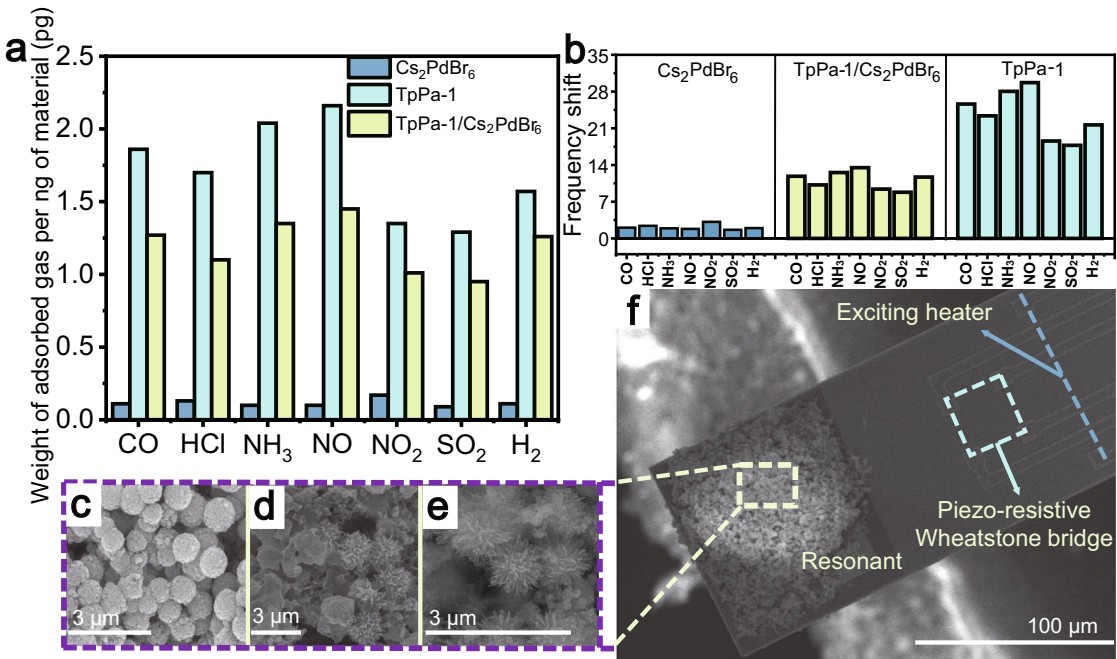

**Fig. 4 | Structure and test performance of the resonant microcantilever. a** The mass of gas adsorbed by 1 ng of material deposited on the microcantilever. **b** Frequency shift of the microcantilever $Cs_2PdBr_6$, TpPa-1/$Cs_2PdBr_6$, and TpPa-1 sensors to various types of gases (all at concentrations of 2 ppm). FSEM images of **c** $Cs_2PdBr_6$, **d** TpPa-1/$Cs_2PdBr_6$, and **e** TpPa-1 deposited on the microcantilever. **f** FSEM image of the structure of the microcantilever. Source data are provided as a Source Data file.

of dioxane (Aladdin, 99% purity) were added to the vial and sonicated for 10 min. Subsequently, the mixture was transferred into a 50 mL Teflon-lined stainless steel autoclave, and 1 mL of 6 M aqueous acetic acid was added and sonicated for 10 min. Finally, the autoclave was heated to 120 °C and kept for 3 days. The product was filtered and washed with N,N-dimethylacetamide (TCI, 99% purity), anhydrous tetrahydrofuran (TCI, 99% purity), and acetone. The collected powder was then dried at 120 °C under vacuum for 12 h to give TpPa-1 COF.

### Synthesis of TpPa-2, TpPa-CN, TpPa-NO₂, and TpPa-COOH
The method is the same as the synthesis of TpPa-1, and only the corresponding raw materials need to be replaced.

### Characterizations of COFs
Five 2D COFs also were synthesized by the Schiff base reactions of 1,3,5-triformylphloroglucinol (Tp) and p-phenylenediamine (Pa-1) (2,5-dimethyl-p-phenylenediamine (Pa-2), 2,5-diaminobenzonitrile (Pa-CN), 2-nitro-1,4-phenylenediamine (Pa-NO₂), and 2,5-diaminobenzoic acid (Pa-COOH), in Supplementary Fig. 3). Typically, to prepare TpPa-1, Tp, and Pa-1 were dissolved in a solvent mixture (mesitylene/dioxane = 1:1) to form a precursor solution, and acetic acid was added as a catalyst. Then, the solution was transferred into the reactor and reacted at 120 °C for 72 h (Supplementary Fig. 3b). TpPa-2, TpPa-CN, TpPa-NO₂, and TpPa-COOH were synthesized under the same conditions (Supplementary Fig. 3c–f). The X-ray diffraction (XRD) pattern confirmed COF's formation as previously reported (Supplementary Fig. 3g–k)[30,42–44]. It is noteworthy that all COFs have a π–π stacking (AA) structure except TpPa-NO₂, which has a staggered (AB) structure. The Brunauer−Emmett−Teller (BET) surface areas of the activated COFs were found to be 59–596 m²/g (Supplementary Fig. 3l–p). The Fourier Transform Infrared (FT-IR) spectra of TpPa-1 indicated total consumption of the starting materials on the basis of the disappearance of the N−H stretching bands of Pa-1 (3100–3300 cm⁻¹) and the carbonyl stretching

bands of Tp (1638 cm⁻¹) (Supplementary Fig. 4a)[30]. The peak positions contributed to the C=C bond and C−N bond in TpPa-1 are shifted compared to the raw materials, which also indicates the successful synthesis of TpPa-1. The FT-IR spectra of several other COFs all have similar shifts (Supplementary Fig. 4b–e).

### Preparation of TpPa-1/Cs₂PdBr₆
160 mg $Cs_2PdBr_6$ powder was dissolved in 1 mL mixed solvent (DMF:DMSO = 1:1) by heating. TpPa-1 (30 mg, 48.4 wt%) was sonicated in 10 mL tert-butanol for 30 min and then stirred for 60 min. Subsequently, slowly drop 200 µL of $Cs_2PdBr_6$ precursor solution into the TpPa-1 suspension and stir for 30 min. The mixture was left to stand for 12 h.

### Preparation of TpPa-1/Cs₂AgBiBr₆, TpPa-1/Cs₃BiBr₉, TpPa-1/Cs₂SnI₆, TpPa-2/Cs₂PdBr₆, TpPa-CN/Cs₂PdBr₆, TpPa-NO₂/Cs₂PdBr₆, and TpPa-COOH/Cs₂PdBr₆
The method is similar to that for the preparation of TpPa-1/$Cs_2PdBr_6$, with only a slight adjustment of the solvent ratio.

### Fabrication of sensor
20 µL TpPa-1/$Cs_2PdBr_6$ precipitation was drop-coated on an $Al_2O_3$ substrate printed with interdigitated electrodes (channel width: 200 µm, MJ-10, Beijing Elite Technology Co. Ltd, China) and dried under an infrared drying lamp.

### Gas sensing measurements
The sensor was placed in a 1200 mL gas chamber, and different concentrations of $NO_2$ gas were introduced into the gas chamber. The sensor current changes were monitored in real-time using the Keithley 4200-SCS. The gas flow rate was always stabilized at 100 mL/min, and the temperature was stabilized at 300 K to reduce the influence of flow rate and temperature on the test. A schematic of the sensing system is presented in Supplementary Fig. 7. The other gas tests are the same as those for $NO_2$.

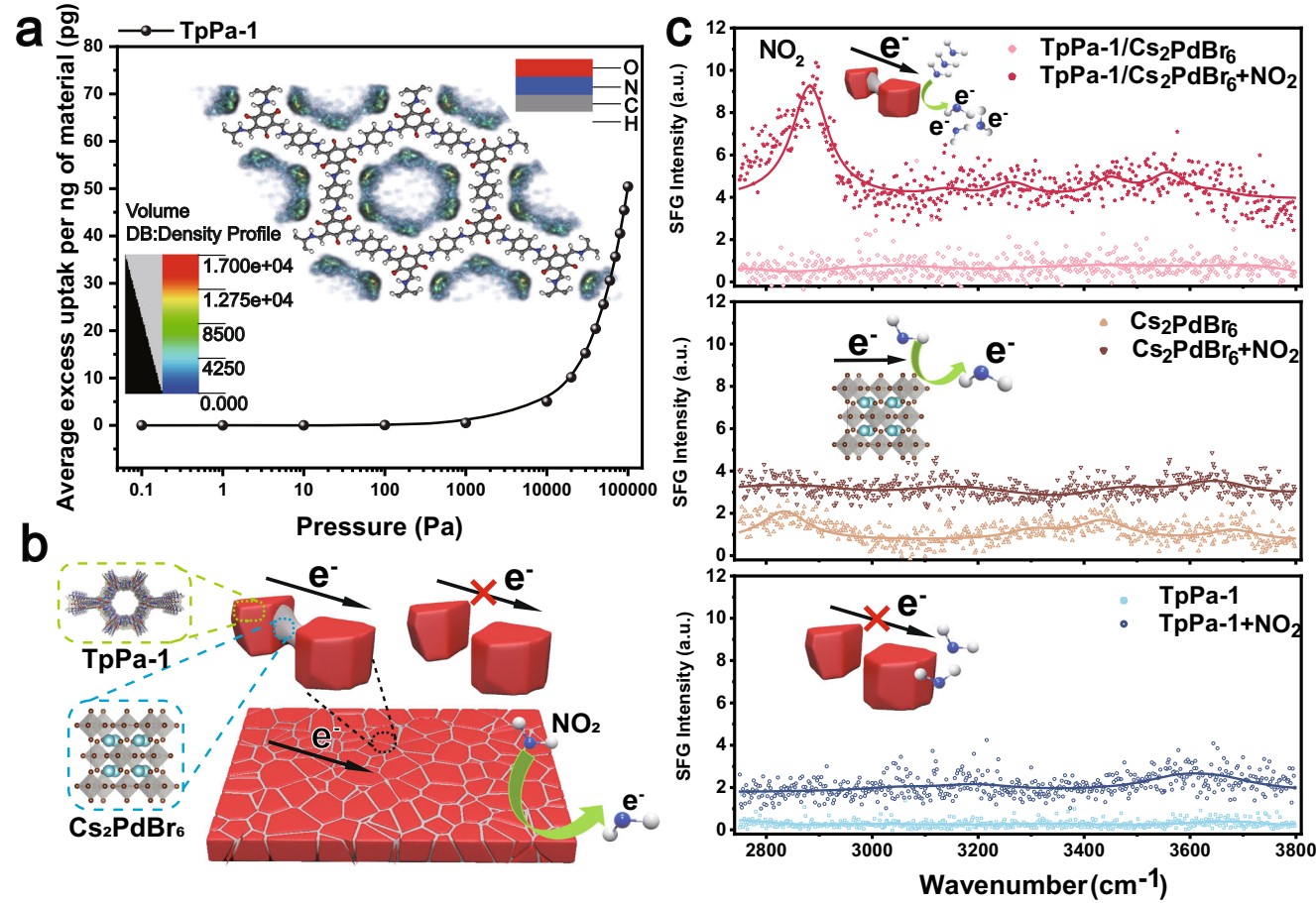

**Fig. 5 | Characterization of the sensing mechanism. a** Grand canonical Monte Carlo simulation of NO$_2$ adsorption density plot on TpPa-1. The insets in **a** show the GCMC simulation of the average excess uptake of the NO$_2$/N$_2$ mixture per ng of TpPa-1. **b** Schematic diagram of the function of electric glue. **c** SFG spectra of the film before and after NO$_2$ adsorption. Source data are provided as a Source Data file.

### Resonant microcantilever fabrication

The length, width, and thickness of the cantilever are 200, 100, and 3 µm, respectively, and its effective mass is about 33 ng. The design and fabrication of the cantilever have been reported in detail[45]. A small amount of COF is ultrasonically dispersed in tert-butanol, and then the suspension is deposited on the cantilever through the sample pre-paration device.

### Resonant microcantilever sensing experiment to gas

Before gas detection, the cantilever is put into a 9.4 mL testing chamber to obtain the baseline signal. When different con-centrations of NO$_2$ are introduced, the material loaded at the free end of the cantilever absorbs the NO$_2$, so that the vibration fre-quency of the cantilever changes. The gas flow rate was always stabilized at 30 mL/min, and the temperature was stabilized at 300 K to reduce the influence of flow rate and temperature on the test. A schematic of the sensing system is presented in Supple-mentary Fig. 15. The frequency change was monitored through frequency−time measurements using intelligent physicochemical parameters analyzer (IPPA). The other gas tests are the same as those for NO$_2$.

### Calculation method for deposited material and adsorbed gas mass on a resonant microcantilever

The resonant microcantilever can convert the mass increase induced by the target analyte molecules' adsorption into a decrease in the resonant frequency of the microcantilever. The mass of the adsorbed gas is proportional to the frequency-shift signal when the gas mass is much smaller than the effective mass of the resonant microcantilever itself.

$$\Delta f \approx 0.5 \Delta m \times \frac{f_0}{m_{\text{eff}}}$$

where $f_0$ is the initial resonant frequency before mass adsorption, $m_{\text{eff}}$ is effective mass of the resonant microcantilever itself. The length, width, and thickness of the cantilever are 200, 100, and 3 µm, respectively. Thus, the effective mass can be calculated as about 33 ng[45,46].

### Measurements and general methods

The microscopic morphologies of all objects were characterized by SEM (HITACHI Japan S-4700) and TEM (FEI TECNAI G20), respectively. The structural characterization of perovskite and COF was determined by XRD (Bruker D8 Advance) and FTIR (VERTEX70). Keithley 4200-SCS was used to test the sensor performance. Resonant microcantilever gas sensing data were recorded using an intelligent physicochemical parameters analyzer (IPPA). The resonant microcantilever is produced by Xiamen High-End MEMS Technology Co., Ltd. To view a copy of this license, visit http://highend-mems.com/product_center. The SFG sys-tem was built by EKSPLA: the visible beam (incident angle 60°, 532 nm) and IR beam (incident angle 55°, around 2700–3800 cm$^{-1}$) were about 25 ps at 50 Hz. Since the energy of visible and IR beams was <200 mJ, the sample photodamage during the test can be ignored.

## Theoretical calculation

All the density functional theory calculations were performed using the Vienna Ab initio Simulation Package (VASP)[47–49]. The exchange and correlation potentials were determined with the Perdew, Burke, and Ernzerhof within the generalized gradient approximation (PBE-GGA) functional[50]. The projector augmented-wave (PAW) method was used to describe the electron wave function[51]. To accurately describe the van der Waals interaction, the DFT-D3 method with Becke–Jonson damping was used in all the calculations[52,53]. The plane wave energy cut-off was set to 520 eV, and the energy convergence was set to $1 \times 10^{-5}$ eV. The lattice supercell (3*3) with a vacuum of 15 Å is composed of the $Cs_2TeI_6$ (111) surface. The geometry optimization is performed when the Hellmann–Feynman force on each atom is under $0.02\ eV\ Å^{-1}$. The crystal orbital Hamilton population (COHP) analysis was performed using LOBSTER code[54]. The optimized structure and the charge density difference distributions were illustrated with VESTA software[55].

## Data availability

The raw data that support the findings of this study are available in https://nomad-lab.eu/prod/v1/gui/search/entries/entry/id/Ym87s8Ebu1txSmdwagUaccQA9Fwc/files/source%20data.xlsx.

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

## Acknowledgements

We gratefully acknowledge the financial support provided by the National Key R&D Program of China (2020YFC1818401, 2017YFC0210906), National Natural Science Foundation of China (21978185, 21938006, 21776190), Basic Research Project of Leading Technology in Jiangsu Province (BK20202012), Suzhou Science and Technology Bureau Project (SYG201935) and the project supported by the Priority Academic Program Development of Jiangsu Higher Education Institutions (PAPD).

## Author contributions

J.-M.L. and J.-H.H. conceived and directed the project. W.Y. carried out key experiments and wrote the manuscript. L.D.Z. and L.F.D. performed grand canonical Monte Carlo simulation calculations. H.-Z.L. performed sum-frequency generation testing and analysis. J.-H.H., W.Y., and Q.C. performed density functional theory theoretical calculations. Z.K.C., J.W., and Q.-M.S. conducted part of the characterizations. W.Y., Q.C., Z.K.C., J.W., Q.-M.S., J.-H.H., and J.-M.L. analyzed the data. All authors discussed the results and commented on the manuscript.

## Competing interests

The authors declare no competing interests.
