## [Peer Review File · Nature Communications]

REVIEWER COMMENTS

Reviewer #1 (Remarks to the Author):

The authors have applied double perovskite Cs₂PdBr₆ nanospheres as electric glues to improve the conductivity of 2D COF crystallite TpPa-1. The as-synthesized complex was used as NO₂ gas sensors which demonstrate low detection limit and high NO₂ selectivity. This work is innovative, but the studies on the mechanism looks not adequate. There are some specific comments for the authors to further improve the manuscript quality, as below;

1. Authors demonstrated that Pd⁴⁺ coordinates with O on the carbonyl group and N on the imine group in TpPa-1, making the perovskite-COF complex, where Pd⁴⁺ are defect-induced species in Cs₂PdBr₆. Authors also studied the universality of electric glue strategy for Cs₂AgBiBr₆, Cs₃Bi₂Br₉ and Cs₂SnI₆. How do those perovskite coordinate with COFs? Is it also defect-related?
2. The mechanism of high NO₂ selectively is not clear. Figure 4b shows the great gas adsorption capacity of porous COFs, but there is no gas selectivity. Why does the sensor show high highest response to NO₂?
3. In Figure 2d, the performance of the TpPa-1/Cs₂PdBr₆ shows degradation after a long time. What results in the degradation? Is it because of the degradation of perovskite? or COFs? or their coordination is broken? Further explanation is needed.
4. On page 10, line 177, the authors explained NO₂ is physically adsorbed on TpPa-1, while NO₂ is also coordinated with Pd on perovskite. It is confusing. Do TpPa-1 and Cs₂PdBr₆ both contribute to the adsorption NO₂? Besides, what effect does the NO₂ orientation have on the charge transport?

Reviewer #2 (Remarks to the Author):

In this research, the authors demonstrated that halide perovskite nanocrystals could be employed as electric glues to bond 2D COF crystallites and significantly improve their conductivity. This as-prepared composites could detect NO₂ with high selectivity and sensitivity. Moreover, resonant microcantilever (RMC), grand canonical Monte Carlo (GCMC), density functional theory (DFT) and sum-frequency generation (SFG) analyses have been employed to prove that 2D COFs could enrich and transfer electrons to NO₂ molecules, leading to increased device conductivity. This work provides an electric "glue" to solve this problem, which is very attractive. I would like to recommend its publication after minor revisions.

1. Are perovskites the only effective electric glue to enhance the conductivity and sensing performance of COFs? What are the advantages of perovskites as electric glue?
2. According my experience, it is difficult for TpPa-1 powder to adhere onto Al₂O₃-based interdigitated electrodes. Can perovskites glue effectively alleviate this problem?
3. Why do Cs₂PdBr₆ exist in the form of nanospheres in the Fig. 1e? According to your previous reports (Adv Mater. 2021, 33, 2100674), Cs₂PdBr₆ should exist in the form of hollow spheres in tert-butanol.
4. The article should give a cross-sectional view of the TpPa-1/Cs₂PdBr₆ film to determine the film thickness. How to control the thickness of TpPa-1/Cs₂PdBr₆ films when fabricating sensing devices?
5. Ambient moisture causes generally serious interference to gas detection. This should be checked.
6. Some papers related to COFs might need to be included in the revised manuscript: 10.1038/S41467-020-19315-6; 10.1002/ANIE.201909613; 10.1002/ADMA.202101175.
7. The method for calculating the mass of adsorbed gas using the resonant microcantilever should be given in the experimental section.
8. Perovskites and COFs still have many issues in the practical application of the sensors. What advantages do they have over traditional metal oxides? What are the next research priorities? The author may comment in the introduction.

Replies to the Reviewers' Comments and Revisions Made to the Manuscript

Reviewer #1 (Remarks to the Author):

The authors have applied double perovskite Cs₂PdBr₆ nanospheres as electric glues to improve the conductivity of 2D COF crystallite TpPa-1. The as-synthesized complex was used as NO₂ gas sensors which demonstrate low detection limit and high NO₂ selectivity. This work is innovative, but the studies on the mechanism looks not adequate. There are some specific comments for the authors to further improve the manuscript quality, as below;

1. Authors demonstrated that Pd⁴⁺ coordinates with O on the carbonyl group and N on the imine group in TpPa-1, making the perovskite-COF complex, where Pd⁴⁺ are defect-induced species in Cs₂PdBr₆. Authors also studied the universality of electric glue strategy for Cs₂AgBiBr₆, Cs₃Bi₂Br₉ and Cs₂SnI₆. How do those perovskite coordinate with COFs? Is it also defect-related?

Reply:

Thanks for your kind comments.

We used XPS to further explore how these perovskites (Cs₂AgBiBr₆, Cs₃Bi₂Br₉ and Cs₂SnI₆) coordinate with TpPa-1. The XPS results show that the exposed metal sites on these perovskites due to halide vacancies coordinate with N and O of TpPa-1. As shown below, the metal ions (Ag, Bi and Sn) peaks in TpPa-1/perovskite blue-shift compared to the pristine perovskite. In addition, both the O and N peaks of TpPa-1 red-shifted after coordination with the perovskites. The XPS result indicates that TpPa-1 and the halide double perovskites form the chelation of metal ions (Ag, Bi and Sn) by O from the carbonyl group and by N on the imine group in TpPa-1.

High-resolution XPS spectra of various perovskites mixed with TpPa-1.

Revision:

Manuscript

Amended in page 6 paragraph 1

We applied this method to other halide double perovskites ($\text{Cs}_2\text{AgBiBr}_6$, $\text{Cs}_3\text{Bi}_2\text{Br}_9$ and Cs_2SnI_6 ; halide double perovskites have better environmental stability than other ABX_3 halide perovskites) and proved good universality for double perovskites (Fig. 3c and Supplementary Fig. 13). The XPS result indicates that TpPa-1 and the halide double perovskites form the chelation of metal ions (Ag, Bi and Sn) by O from the carbonyl group and by N on the imine group in TpPa-1 (Supplementary Fig. 14).

Supporting information

Added Supplementary Fig. 14

Supplementary Fig. 14: High-resolution XPS spectra of various perovskites mixed with TpPa-1.

2. The mechanism of high NO₂ selectively is not clear. Figure 4b shows the great gas adsorption capacity of porous COFs, but there is no gas selectivity. Why does the sensor show high highest response to NO₂?

Reply:

Thanks for your kind comments.

TpPa-1 has good selectivity for NO₂ sensing (Supplementary Fig. 12), which is further enhanced by the addition of perovskite glue.

In principle, the response of a chemresistor to gaseous molecules (the change of device's conductivity) is determined by the amount of adsorbed gas and the charge transfer degree between sensory materials and gaseous molecules (Adv Mater. 2017, 29, 1703192), which will modulate the film carrier concentration. However, the resonance microcantilever (RMC) test and the grand canonical Monte Carlo (GCMC) simulation have demonstrated that the adsorption capacity for NO₂ on COF is the not the largest, smaller than that of CO, NO, NH₃ and H₂. (Fig. 4a and 5a). Our DFT calculation results show that NO₂ is adsorbed on TpPa-1 and the sum frequency

generation (SFG) signal intensity increase of the TpPa-1 film after treatment with NO₂, indicating that significant charge transfer occurs after NO₂ adsorption (Fig. 5c). Many previous studies also demonstrate that NO₂ always has the strongest charge transfer with organic semiconductor film (Angew Chem, Int Ed, 2021, 60, 15328; J. Am. Chem. Soc. 2019, 141, 11929). Therefore, the selectivity of TpPa-1 for NO₂ is mainly determined by the charge transfer.

Revision:

Manuscript

Amended in page 9 paragraph 2

As shown in Fig. 5c, the SFG signal intensity of the three types of membranes significantly increases after treatment with NO₂, indicating that charge transfer occurs after NO₂ adsorption. The TpPa-1/Cs₂PdBr₆ film showed the greatest enhancement of the background signal, which indicates that NO₂ produces a stronger charge transfer between NO₂ and TpPa-1/Cs₂PdBr₆.

In addition, when the NO₂ molecule is close to TpPa-1/Cs₂PdBr₆, it will coordinate with Pd on the perovskites (Supplementary Fig. 24). Therefore, the TpPa-1/Cs₂PdBr₆ film present a distinct characteristic peak of NO₂ after adsorption, which indicates NO₂ has a relatively high orientation uniformity and selectivity at room temperature. (Fig. 5c).^{37,38} Our analysis shows that charge transfer between TpPa-1 and NO₂ molecules and the bond of Cs₂PdBr₆ glue to TpPa-1 crystallites significantly increase the base conductivity of the TpPa-1 film, which gives the sensor excellent sensing performance.

3. In Figure 2d, the performance of the TpPa-1/Cs₂PdBr₆ shows degradation after a long time. What results in the degradation? Is it because of the degradation of perovskite? or COFs? or their coordination is broken? Further explanation is needed.

Reply:

Thanks for your kind comments.

The notorious decomposition of perovskite when exposed to atmosphere with oxygen and particularly moisture, leads to the degradation of device performance. (Adv Mater. 2019, 31, 1805337.) To figure out the reason for the degradation of the sensor performance, we analyzed the device placed in the atmosphere for 300 days. The XRD results show that partial decomposition of the Cs_2PdBr_6 occurred after the sensor is placed for 300 days, but the TpPa-1 is still stable (Figure a). XPS revealed that Pd undergoes a clear valence change (Figure b). In addition, only N is involved in the coordination (Figure c and d), which may be the coordination structure change resulting from the decomposition of Cs_2PdBr_6 . In order to further improve the stability of the sensor and enhance the practical ability of the sensor, the sensors need to work with a dehumidification system such as a filter.

Changes in the XRD pattern and XPS spectra of the TpPa-1/Cs₂PdBr₆ after the sensor was placed in the atmosphere for 300 days.

4. On page 10, line 177, the authors explained NO₂ is physically adsorbed on TpPa-1, while NO₂ is also coordinated with Pd on perovskite. It is confusing. Do TpPa-1 and Cs₂PdBr₆ both contribute to the adsorption NO₂? Besides, what effect does the NO₂ orientation have on the charge transport?

Reply:

Thanks for your kind comments.

- (1) Yes, both TpPa-1 and Cs₂PdBr₆ contribute to NO₂ adsorption as demonstrated by resonance microcantilever (RMC) test (Fig. 4b). Gas molecules will be anchored by coordination or electrostatic adsorption at the Br⁻ vacancy of Cs₂PdBr₆. However, the selectivity of Cs₂PdBr₆ to adsorbed gas is weak, which has been stated in the article (Fig. 3b and S12a) and our previous work (Adv Mater. 2021, 33, 2100674).
- (2) The orientation of NO₂ does not directly affect the charge transport, but it determines the charge transfer between NO₂ and the sensory film, in consequence affect the charge transport of the film. As we shown in figure below, we obtained the optimal adsorption site of NO₂ on COF and differential charge density of NO₂ by DFT calculation. The yellow shading represents the electron gaining and the blue shading represents the charge withdrawn. If the orientation of NO₂ changes, the optimal NO₂ adsorption configuration will be destroyed, thereby reducing the transfer of charges.

The charge transfer difference between NO₂ and TpPa-1 layers.

Reviewer #2 (Remarks to the Author):

In this research, the authors demonstrated that halide perovskite nanocrystals could be employed as electric glues to bond 2D COF crystallites and significantly improve their conductivity. This as-prepared composites could detect NO₂ with high selectivity and sensitivity. Moreover, resonant microcantilever (RMC), grand canonical Monte Carlo (GCMC), density functional theory (DFT) and sum-frequency generation (SFG) analyses have been employed to prove that 2D COFs could enrich and transfer electrons to NO₂ molecules, leading to increased device conductivity. This work provides an electric “glue” to solve this problem, which is very attractive. I would like to recommend its publication after minor revisions.

1. Are perovskites the only effective electric glue to enhance the conductivity and sensing performance of COFs? What are the advantages of perovskites as electric glue?

Reply:

Thanks for your kind comments.

(1) Perovskites are not the only electric glue to improve COFs sensing performance.

But perovskites are certainly one of the excellent candidates for electric glue.

(2) Perovskites electric glue have both suitable conductivity and viscosity. In order to highlight the dual advantages of perovskites, we added carbon black and iodine as control additives in the same molar amount as Cs₂PdBr₆ into TpPa-1, respectively.

The sensors were prepared in the same procedure. **Figure a** shows that the sensor's response is significantly reduced due to the high conductivity of carbon black.

Figure b shows that the sensor with added iodine has excellent NO₂ sensing performance, but it is still slightly inferior to the TpPa-1/Cs₂PdBr₆ sensor (lowest detection limit reach 40 ppb). In addition, the Cs₂PdBr₆ can well bond the TpPa-1 powder to form a thin film on the interdigital electrode, but the TpPa-1 film doped with carbon black and iodine has a little peeling off (**Figure c**).

Responses of (a) TpPa-1/C and (b)TpPa-1/I₂ sensors to different NO₂ concentrations. (c) Photographs of sensors based on different materials.

2. According to my experience, it is difficult for TpPa-1 powder to adhere onto Al₂O₃-based interdigitated electrodes. Can perovskites glue effectively alleviate this problem?

Reply:

Thanks for your kind comments.

After adding perovskite, the film can be stably attached to Al₂O₃-based interdigitated electrodes. As shown in the figure below, the films deposited with TpPa-1 alone are prone to fall off, and a stable film can be formed with the addition of a certain amount of Cs₂PdBr₆ under the same preparation conditions.

Photographs of sensors based on different materials.

3. Why do Cs_2PdBr_6 exist in the form of nanospheres in the Fig. 1e? According to your previous reports (Adv Mater. 2021, 33, 2100674), Cs_2PdBr_6 should exist in the form of hollow spheres in tert-butanol.

Reply:

Thanks for your kind comments.

High-speed stirring and sonication during TpPa-1/ Cs_2PdBr_6 preparation will destroy the structure of perovskite hollow spheres. We prepared Cs_2PdBr_6 hollow spheres under low-speed (500 rpm) stirring and ultrasonic environments, respectively. As shown in the figure below, the ultrasonic dispersion destroys the hollow sphere structure of the Cs_2PdBr_6 . In addition, TpPa-1 dispersed in tert-butanol may also affect the formation of Cs_2PdBr_6 hollow spheres.

Perovskite hollow spheres grown under (a) low-speed (500 rpm) stirring and (b) sonication.

4. The article should give a cross-sectional view of the TpPa-1/ Cs_2PdBr_6 film to determine the film thickness. How to control the thickness of TpPa-1/ Cs_2PdBr_6 films when fabricating sensing devices?

Reply:

Thanks for your kind comments.

- (1) We have made corresponding replenish in the supporting information.
- (2) We used the drop coating method to fabricate the gas sensor. 20 μL TpPa-1/ Cs_2PdBr_6 precipitation was drop-coated on an Al_2O_3 substrate printed with interdigitated electrodes and dried under an infrared drying lamp. Therefore, it can be ensured that

the thickness of TpPa-1/Cs₂PdBr₆ film remains uniform.

Revision:

Manuscript

Amended in page 14 paragraph 4

20 μ L TpPa-1/Cs₂PdBr₆ precipitation was drop-coated on an Al₂O₃ substrate printed with interdigitated electrodes (channel width: 200 μ m, MJ-10, Beijing Elite Technology Co. Ltd, China) and dried under an infrared drying lamp.

Supporting information

Added Supplementary Fig. 6a

Supplementary Fig. 6: **a** Cross-sectional SEM images of device structure (TpPa-1/Cs₂PdBr₆ film \approx 100 μ m). **b** The change of current with time when the TpPa-1 and TpPa-1/Cs₂PdBr₆ sensor is exposed to N₂ (99.999%). A constant voltage of 5 V was applied across the interdigital electrodes. **c** Specifications of interdigitated electrodes.

5. Ambient moisture causes generally serious interference to gas detection. This should be checked.

Reply:

Thanks for your kind comments.

The sensor can maintain good sensing performance in different humidity environments. At a temperature of 300K, we tested the sensing performance of NO₂ in a humidity environment of 11% - 90%RH. The graph below shows that the conductivity of the material increases with increasing humidity, but the sensing performance is not greatly affected. We can remove the effect of humidity by baseline calibration or adding

dehumidification equipment.

Responses variation versus NO₂ concentrations (40 ppb to 10 ppm) under different humidity conditions.

Revision:

Manuscript

Amended in page 4 paragraph 2

The TpPa-1/Cs₂PdBr₆ exhibit superiority in chemresistive detection of NO₂ with a detection limit of 40 ppb, which is lower than that of most NO₂ sensors (Supplementary Fig. 9). Even under atmosphere with different humidity, the sensor can still work smoothly (Supplementary Fig. 10).

Supporting information

Added Supplementary Fig. 9

Supplementary Fig. 9: Responses variation versus NO₂ concentrations (40 ppb to 10 ppm) under different humidity conditions.

6. Some papers related to COFs might need to be included in the revised manuscript:
10.1038/S41467-020-19315-6; 10.1002/ANIE.201909613;
10.1002/ADMA.202101175.

Reply:

Thank you for your suggestions. We have added these references in the manuscript.

Revision:

Manuscript

Amended in page 2 paragraph 1

Two-dimensional covalent organic frameworks (2D COFs) are promising for resistance-related applications, such as gas sensing, optoelectronic devices and photoelectric catalysis, owing to their semiconducting nature, large surface area, size-tunable pores and abundant active sites.^{1, 2, 3, 4, 5} Among them, gas sensing plays an indispensable role in various fields, such as food quality assessment, plant growth detection, and noninvasive medical diagnosis.

References

3. Yu F, Liu W, Li B, Tian D, Zuo JL, Zhang Q. Photostimulus - responsive large - area two - dimensional covalent organic framework films. *Angew Chem, Int Ed* **58**, 16101-16104 (2019).
4. She P, Qin Y, Wang X, Zhang Q. Recent progress in external - stimulus - responsive 2D covalent organic frameworks. *Adv Mater* **34**, 2101175 (2022).
5. Yu F, Liu W, Ke S-W, Kurmoo M, Zuo J-L, Zhang Q. Electrochromic two-dimensional covalent organic framework with a reversible dark-to-transparent switch. *Nat Commun* **11**, 1-6 (2020).

7. The method for calculating the mass of adsorbed gas using the resonant microcantilever should be given in the experimental section.

Reply:

Thank you for your suggestions. We have added the method for calculating the mass of adsorbed gas using the resonant microcantilever in the manuscript.

Revision:

Manuscript

Amended in page 15 paragraph 4

Calculation method for deposited material and adsorbed gas mass on a Resonant microcantilever

The resonant microcantilever can convert the mass increase induced by the target analyte molecules adsorption into the decrease of the resonant frequency of the microcantilever. The mass of the adsorbed gas is proportional to the frequency-shift signal when the gas mass is much smaller than effective mass of the resonant microcantilever itself.

$$\Delta f \approx 0.5\Delta m \times \frac{f_0}{m_{\text{eff}}}$$

where f_0 is the initial resonant frequency before mass adsorption, m_{eff} is effective mass of the resonant microcantilever itself. The length, width, and thickness of the cantilever are 200, 100, and 3 μm , respectively. Thus, the effective mass can be calculated as about 33 ng.^{43,44}

8. Perovskites and COFs still have many issues in the practical application of the sensors. What advantages do they have over traditional metal oxides? What are the next research priorities? The author may comment in the introduction.

Reply:

Thanks for your kind comments.

(1) Halide perovskites as sensing materials have the following advantages: a. Halide perovskites are simple to synthesize and can be prepared quickly. b. Halide perovskites are easily soluble and can form films on various substrate surfaces, which is beneficial to the miniaturization, integration and flexibility of sensor devices. c. Halide perovskites can be used to design sensing materials with different functions by tuning the structure and elemental composition. d. Halide perovskite sensing devices can detect at room temperature, reducing energy consumption. Covalent-organic frameworks (COFs) as sensing materials have the following

advantages: COFs have the characteristics of predictable structure, adjustable structure, ease of functionalization and high specific surface area. The COFs can be designed according to the target gas.

- (2) Future research directions will focus on selectivity, stability, and conductivity. Halide perovskites still suffer from poor selectivity and stability as sensing materials. The selectivity of halide perovskites can be improved by adding a gas separation layer or modifying and passivating perovskites. The stability can be improved by designing the structure of perovskites, such as double perovskites and 2D layered perovskites. COFs have problems such as poor conductivity and difficulty in film formation. The conductivity of COFs can be improved by means of molecular design and doping. COFs can be grown on substrates by conventional methods such as growth and electrodeposition.

Revision:

Manuscript

Amended in page 2 paragraph 1

Therefore, a great deal of effort has been devoted to tuning the conductivity of 2D COFs through molecular formula innovation, including monomers, linkages, and defects.¹⁰

However, 2D COFs usually grow in the form of insoluble polycrystalline powders, where the poor contacts in grain boundaries severely inhibit the macroscopic conductivity of COFs.⁹ Guest molecule doping aims to increase the carrier concentration in COF molecules within grains, the access to COFs with high bulk conductivity remains limited, and the manufacturing process is difficult.^{11, 12} Alternately, the preparation of continuous films or single crystals of COFs is also rather challenging and consequently cost-ineffective for potentially scalable applications.^{13, 14, 15, 16} Therefore, it is more practical to find an effective technique that can improve the contact resistances of COF crystallites for direct integration into sensors.

Reviewer #3 (Remarks to the Author):

The author has demonstrated that halide perovskite nanocrystals can be employed as electric glues to bond 2D covalent organic frameworks (COF) crystallites to significantly improve their conductivity by two orders of magnitude, activating them to detect NO₂ with high selectivity and sensitivity. Resonant microcantilever (RMC), grand canonical Monte Carlo (GCMC), density functional theory (DFT) and sum-frequency generation (SFG) analyses prove that 2D COFs can enrich and transfer electrons to NO₂ molecules, leading to increased device conductivity. I find that the manuscript is well-suited for publication. However, I find that the manuscript is not acceptable in the present form and recommend the manuscript for publication once the concerns listed below are addressed.

Comments

1. In the title, better to add NO₂ sensing instead of using the general term selective gas sensing.

Reply:

Thank you for your suggestions. We have made changes to the article title.

Revision:

Manuscript

Amended in page 1 paragraph 1

Halide Perovskite Glues Activate Two-dimensional Covalent Organic Framework Crystallites for Selective NO₂ Sensing

2. In the abstract, the authors mentioned “applications”. I recommend being specific, which applications?

Reply:

Thank you for your suggestions. We have made changes in the abstract.

Revision:

Manuscript

Amended in page 1 paragraph 4

Resonant microcantilever, grand canonical Monte Carlo, density functional theory and sum-frequency generation analyses prove that 2D COFs can enrich and transfer electrons to NO₂ molecules, leading to increased device conductivity. This work provides a facile approach to improving the conductivity of polycrystalline 2D COF films and may expand their applications in semiconductor devices, such as sensors, resistors, memristors and field-emission transistors.

3. No need to abbreviate more than one time. RMC, GCMC, DFT, and SFG are abbreviated twice (see the last paragraph of the introduction).

Reply:

Thank you for your suggestions. We have removed redundant full names in the manuscript.

4. Which other 20 gases are engaged here? Please mention it at least once in the manuscript.

Reply:

Thank you for your suggestions. We have made changes in the manuscript.

Revision:

Manuscript

Amended in page 5 paragraph 1

A high selectivity of the sensor's response to NO₂ is 70 times or more sensitive than that to the other 20 gases (CO, HCl, NH₃, NO, SO₂, H₂, acetone, 3-pentanone, ethyl acetate, butyl acetate, toluene, chlorobenzene, benzaldehyde, anisole, isopropanol, ethanol, n-heptane, n-hexane, acetic acid, and formic acid. Fig. 2c).

5. Give details or cite a reference for the “antisolvent growth method”.

Reply:

Thank you for your suggestions. We have given the experimental details of the "antisolvent growth method " in the manuscript.

Revision:

Manuscript

Amended in page 3 paragraph 2

In addition, we successfully used perovskite nanospheres to link 2D COF crystallites by an antisolvent growth method. First, TpPa-1 powders are dispersed in tert-butanol by sonification, and then the Cs₂PdBr₆ solution is added dropwise to the dispersion and fully stirred to form TpPa-1/Cs₂PdBr₆ (Supplementary Fig. 5).

Manuscript

Amended in page 14 paragraph 2

Preparation of TpPa-1/Cs₂PdBr₆. 160 mg Cs₂PdBr₆ powder was dissolved in 1 mL mixed solvent (DMF: DMSO=1:1) by heating. TpPa-1 (30 mg, 48.4% wt) was sonicated in 10 mL tert-butanol for 30 minutes and then stirred for 60 minutes. Subsequently, slowly drop 200 μL of Cs₂PdBr₆ precursor solution into the TpPa-1 suspension and stir for 30min. The mixture was left to stand for 12h.

6. The authors found, “We placed the sensor in the atmosphere for 160 days, and the sensor still had a good response to different concentrations of NO₂ (Fig. 2d).” What parameters are involved to decide “good response” and how good? Can authors talk in terms of numbers/percentages etc.?

Reply:

Thank you for your suggestions. We have made changes in the manuscript.

Revision:

Manuscript

Amended in page 5 paragraph 1

Stability is also an important parameter of sensor performance. After 160 days in the atmosphere, the sensor is still able to distinguish between different NO₂ concentrations with a response greater than 60 at 2 ppm of NO₂ (Fig. 2d).

7. The author found, “The results showed that, with the decrease in the amount of doping, the response and selectivity to NO₂ showed a trend of first increasing and then decreasing ...”. What is the scientific reason behind this trend?

Reply:

Thank you for your suggestions.

The power-law response to polymers toward oxidants can be modelled it using a resistors-in-series mode as follows: $S = \frac{I-I_0}{I_0} \propto \theta$ Where S is the response, and θ is the surface adsorbate coverage (number of adsorbed species on the surface/number of adsorbing available sites, J. Electrochem. Soc.1999, 146, 1231). The COFs reveal more active sites as the content of perovskite glue declines, which enhances the sensor's responsiveness and selectivity. If the amount of perovskite glue keeps dropping, the conductivity of COFs will diminish quickly and the interaction of NO₂ molecules with COFs won't result in a significant change in current that would affect the sensor's response. Therefore, with the decrease in the amount of doping, the response and

selectivity to NO₂ showed a trend of first increasing and then decreasing.

8. The authors found, “The DFT calculation results show that NO₂ is physically adsorbed on TpPa-1 (Supplementary Fig. 18)”. The mentioned figure contains charge density difference which is not a parameter to determine chemical or physical absorption instead Electron localization function (ELF) can do that. See reference J. Phys.: Condens. Matter 32 (2020) 315502 (12pp) <https://doi.org/10.1088/1361-648X/ab7fd8>

Reply:

Thank you for your suggestions. We have made changes in the manuscript.

The 2D electron localization function (ELF) of NO₂ adsorbed (a) on the surface and (b) within the pores of monolayer TpPa-1.

Revision:

Manuscript

Amended in page 9 paragraph 2

The DFT calculation results show that NO₂ is physically adsorbed on TpPa-1 (Supplementary Fig. 22 and 23).

Supporting information

Added Supplementary Fig. 22

Supplementary Fig. 22: The 2D electron localization function (ELF) of NO₂ adsorbed (a) on the surface and (b) within the pores of monolayer TpPa-1. The yellow, pink, red and green balls refer to C, H, O and N atoms, respectively.

9. How many sites were taken into consideration in the DFT approach? Give binding energies of each case to figure out the ground state absorption. Such information will further help to know the absorption strength. Compare the binding energy of NO₂ with monolayer PtSe₂ as this material has already been claimed superior for sensing NO₂ with the ideal binding energy of -0.432 eV (DOI: 10.1002/admi.201600911).

Reply:

Thank you for your suggestions.

(1) In our calculation, we initially propose twelve adsorption models as listed below, based on the fact that 2D COFs have layered and porous structures and that NO₂ molecules typically adsorb in the pores and surface. The adsorption energy is determined by $E_a = E_{(\text{TpPa-1} + \text{NO}_2)} - E_{(\text{TpPa-1})} - E_{(\text{NO}_2)}$, where the individual terms represent the total energies of monolayer TpPa-1 with adsorbed NO₂, pristine monolayer TpPa-1, and free NO₂, respectively. Negative values of E_a reflect exothermic adsorption.

(2) The geometry in Supplementary Fig. 23 has the largest adsorption energy and the most stable structure, as indicated in the figure below and table S1. The binding energy NO₂ on TpPa-1 (0.326 eV) is lower than on PtSe₂ (0.432 eV), allowing for a quicker recovery of the sensor signal (Figure 2a in the references. ACS Nano 2016, 10, 9550). Experimentally, many factors contribute to the sensitivity, including the morphology (microstructure), grain size, humidity, no. of activated adsorption sites, diffusion of gas, effect of defects etc (Materials Today: Proceedings, 2022, 49, 3245). Thanks to the higher surface areas of 2D COF and synergetic effect of perovskite gules and 2D COF, our sensors have a much higher sensitivity than PtSe₂.

Side and top views of the adsorption configurations for NO₂ (a) on the surface and (b) within the pores of monolayer TpPa-1. The yellow, pink, red and green balls refer to C, H, O and N atoms, respectively.

Table S1. Total energy and adsorption energy of the corresponding adsorption structures as shown in figure. $E_a = E_{(\text{TpPa-1} + \text{NO}_2)} - E_{(\text{TpPa-1})} - E_{(\text{NO}_2)}$, $E_{\text{TpPa-1}} = -507.662$ eV, $E_{\text{NO}_2} = -507.18.093$ eV

	I	II	III	IV	V	VI
Total energy E (eV)	-526.081	-526.015	-526.047	-526.070	-526.035	-526.052
Adsorption energy E_a (eV)	-0.326	-0.259	-0.292	-0.314	-0.280	-0.297
	VII	VIII	IX	X	XI	XII
Total energy E (eV)	-526.042	-526.011	-526.025	-526.037	-526.015	-526.027
Adsorption energy E_a (eV)	-0.287	-0.256	-0.269	-0.281	-0.259	-0.271

Manuscript

Amended in page 9 paragraph 2

Therefore, the TpPa-1/Cs₂PdBr₆ film has an excellent response to NO₂. We initially propose twelve adsorption models based on the fact that 2D COFs have layered and porous structures and that NO₂ molecules typically adsorb on the surface and within the pores (Supplementary Fig. 21 and Table S1).

Supporting information

Added Supplementary Fig. 21

Supplementary Fig. 21: Side and top views of the adsorption configurations for NO₂ (a) on the surface and (b) within the pores of monolayer TpPa-1. The yellow, pink, red and green balls refer to C, H, O and N atoms, respectively.

Supporting information

Added Table S1

Table S1. Total energy and adsorption energy of the corresponding adsorption structures as shown in figure. $E_a = E_{(TpPa-1 + NO_2)} - E_{(TpPa-1)} - E_{(NO_2)}$, $E_{TpPa-1} = -507.662$ eV, $E_{NO_2} = -507.18.093$ eV

	I	II	III	IV	V	VI
Total energy E (eV)	-526.081	-526.015	-526.047	-526.070	-526.035	-526.052
Adsorption energy E_a (eV)	-0.326	-0.259	-0.292	-0.314	-0.280	-0.297
	VII	VIII	IX	X	XI	XII
Total energy E (eV)	-526.042	-526.011	-526.025	-526.037	-526.015	-526.027
Adsorption energy E_a (eV)	-0.287	-0.256	-0.269	-0.281	-0.259	-0.271

10. Give the binding energies of the other 20 gases. It will help to get to know the overall behavior of COF towards gas sensing.

Reply:

Your suggestion is valid. It aids in understanding TpPa-1's general behavior for gas sensing. We carefully considered the finances and resources needed to finish this computation and determined that it is not currently feasible to conduct such an enlarged study. Meanwhile, we feel that the scope of work of the present paper can support its conclusions. As a result, we suggest that the additional theoretical calculation is included in a follow-up paper.

11. In the conclusion, the authors claimed, “Overall, this work takes a different approach and directs the path to ...”. What is the difference between the current approach and the already existing approaches? The authors should draw a brief comparison at least to highlight the practicality and advantages of their approach over the commonly employed methodologies.

Reply:

Thanks for your kind comments. We have made supplements in the manuscript.

Revision:

Manuscript

Amended in page 2 paragraph 2

However, 2D COFs usually grow in the form of insoluble polycrystalline powders, where the poor contacts in grain boundaries severely inhibit the macroscopic conductivity of COFs.⁹ Guest molecule doping aims to increase the carrier concentration in COF molecules within grains, the access to COFs with high bulk conductivity remains limited, and the manufacturing process is difficult.^{11, 12}

Manuscript

Amended in page 3 paragraph 1

Halide perovskites lower the boundary resistance between the COF crystallites without short-circuiting, in sequence modulating the entire sensor's basis conductance suitable for gas sensing. Our work gives a simple and effective approach to improve the conductivity of 2D COF films and expand the application of 2D COF crystallites in resistance-related applications.

Manuscript

Amended in page 12 paragraph 1

The TpPa-1/Cs₂PdBr₆ sensor realized high selectivity (the sensor's response to NO₂ is 70 times or more sensitive than that to the other 20 gases) and high sensitivity (the lowest detection limit can reach 40 ppb) in the detection of NO₂. Overall, this work takes a simple and effective approach and has important implications for improving the conductivity of 2D COFs and advancing their resistance-related applications.

References

11. Mulzer CR, *et al.* Superior charge storage and power density of a conducting polymer-modified covalent organic framework. *ACS Cent Sci* **2**, 667-673 (2016).

12. Wu Y, Yan D, Zhang Z, Matsushita MM, Awaga K. Electron highways into nanochannels of covalent organic frameworks for high electrical conductivity and energy storage. *ACS Appl Mater Interfaces* **11**, 7661-7665 (2019).

12. How the charge transfer is calculated. Give the details in the methodology section. What was the amount of charge transfer?

Reply:

Thank you for your suggestions. The conclusion that charge transfer between Cs₂PdBr₆/TpPa-1 and NO₂ is not derived from theoretical calculations, but from SFG test results. Due to the work function difference between Cs₂PdBr₆/TpPa-1 and NO₂, a charge transfer occurred at the contact interface of Cs₂PdBr₆/TpPa-1 and NO₂ which induced a built-in electric field between Cs₂PdBr₆/TpPa-1 and NO₂. The magnitude of the SFG background signal reflected the increase in this built-in electric field and charge transfer (*Adv Mater*, 2021, 33, 2100674).

13. I strongly recommend commenting on the reusability of the sensor. If the binding energy is very high then how feasible will it be to make the sensor available for next time use.

Reply:

(1) Thank you for your suggestions. The binding energies of NO₂ on the surface and within the pores of 2D COF are 0.326 and 0.287 eV, respectively, as shown in supplemental fig.21 and table S1, which are not particularly high when compared to PtSe₂. As you suggest, we evaluated the reusability of the sensor and amend the result as Supplementary Fig. 10. We consistently evaluated the sensor's performance using the same device (see the Methods for details). As shown below, the device maintains a relatively stable working state for four gas sensing tests. Under a constant bias voltage of 5 V, the sensor responds to the flushing of NO₂

flow with increasing concentration.

(2) Heating the sample and purging with a carrier gas can enhance the recovery performance of sensory materials with high binding energies to gases.

Responses variation versus NO₂ concentrations (40 ppb to 10 ppm) for four times gas sensing test.

Revision:

Manuscript

Amended in page 4 paragraph 2

Multiple tests of the same device prove the sensor's excellent reusability (Supplementary Fig. 10). At a concentration of 10 ppm NO₂, the response/recovery time was 71/254 s (Supplementary Fig. 11).

Supporting information

Added Supplementary Fig. 10

Supplementary Fig. 10: Responses variation versus NO_2 concentrations (40 ppb to 10 ppm) for four times gas sensing test.

REVIEWERS' COMMENTS

Reviewer #2 (Remarks to the Author):

The authors have addressed my comments carefully and correctly. It can be accepted now

Reviewer #3 (Remarks to the Author):

Thank you for addressing all my concerns. The manuscript is in the right form of acceptance and I recommend its publication in the journal.